# Unlocking the Power of Parallel Computing: GPU technologies for Ocean Forecasting

Andrew R. Porter[1] and Patrick Heimbach[2]

[1]Science and Technology Facilities Council, Daresbury Laboratory, Hartree Centre, Daresbury, UK
[2]Oden Institute for Computational Engineering and Sciences, The University of Texas at Austin, USA

*Correspondence to*: Andrew Porter (andrew.porter@stfc.ac.uk)

**Abstract.** Operational ocean forecasting systems are complex engines that must execute ocean models with high performance to provide timely products and datasets. Significant computational resources are then needed to run high-fidelity models and, historically, technological evolution of microprocessors has constrained data parallel scientific computation. Today, GPUs offer a rapidly growing and valuable source of computing power rivaling the traditional CPU-based machines: the exploitation of thousands of threads can significantly accelerate the execution of many models, ranging from traditional HPC workloads of finite-difference/volume/element modelling through to the training of deep neural networks used in machine learning and artificial intelligence. Despite the advantages, GPU usage in ocean forecasting is still limited due to the legacy of CPU-based model implementations and the intrinsic complexity of porting core models to GPU architectures. This review explores the potential use of GPU in ocean forecasting and how the computational characteristics of ocean models can influence the suitability of GPU architectures for the execution of the overall value chain: it discusses the current approaches to code (and performance) portability, from CPU to GPU, including tools that perform code-transformation, easing the adaptation of Fortran code for GPU execution (like PSyclone), direct use of OpenACC directives (like ICON-O), adoption of specific frameworks that facilitate the management of parallel execution across different architectures, and also the use of new programming languages and paradigms

## 1 Introduction

Operational Ocean Forecasting Systems (OOFS) are computationally demanding, and large compute resources are required in order to run models of useful fidelity. However, this is a time of great upheaval in the development of computer architectures. The ever-shrinking size of transistors means that current leakage (and the resulting heat generated) now presents a significant challenge to chip designers. This breakdown of 'Dennard Scaling' (transistor power consumption is proportional to area as in Dennard et al., 1974) began in about 2006 and means that it is no longer straightforward to continually increase the clock frequency of processors. Historically this has been the main source of performance

improvement from one generation of processor to the next (Figure 1). Although the number of transistors per device continues to rise, they are increasingly being used to implement larger numbers of execution cores. It is then the job of the application to make use of these additional cores to achieve a performance improvement. Graphical Processing Units (GPUs) are a natural consequence of this evolution. Originally developed to accelerate rendering of computer-generated images (a naturally data-parallel task thanks to the division of an image into pixels), scientists were quick to seize on their potential to accelerate data-parallel scientific computation. Therefore, manufacturers today produce HPC-specific "GPUs" that are purely intended for computation. The suitability of this hardware for the training of deep neural networks used in machine learning and artificial intelligence has stimulated massive development and competition amongst GPU vendors. Because of the exploding interest of AI applications in virtually all sectors of industry, the commercial HPC market is undergoing a seismic shift toward GPU-based hardware, with serious implications for available HPC architectures in the future, to which OOPC will have to adapt.

Unlike CPUs which tend to have relatively few but powerful (general purpose) processor cores, GPUs support hundreds of simpler cores running thousands of threads which can get data from memory very efficiently. The simplicity of these cores makes them more energy efficient and therefore GPUs tend to offer significantly greater performance per Watt. With energy consumption of large computing facilities now the key design criterion, GPUs are an important part of the technology being used in the push towards Exascale performance and beyond (e.g. Draeger and Siegel, 2023). As an illustration, in the November 2024 incarnation of the Top500 list (Strohmaier et al., 2024), nine of the machines in the top ten are equipped with GPU accelerators from NVIDIA, Intel or AMD. Although CPUs are present in these machines, their primary role is to host the GPUs which provide the bulk of the compute performance. GPUs are therefore a major feature of the current HPC landscape, and their importance and pervasiveness is only set to increase.

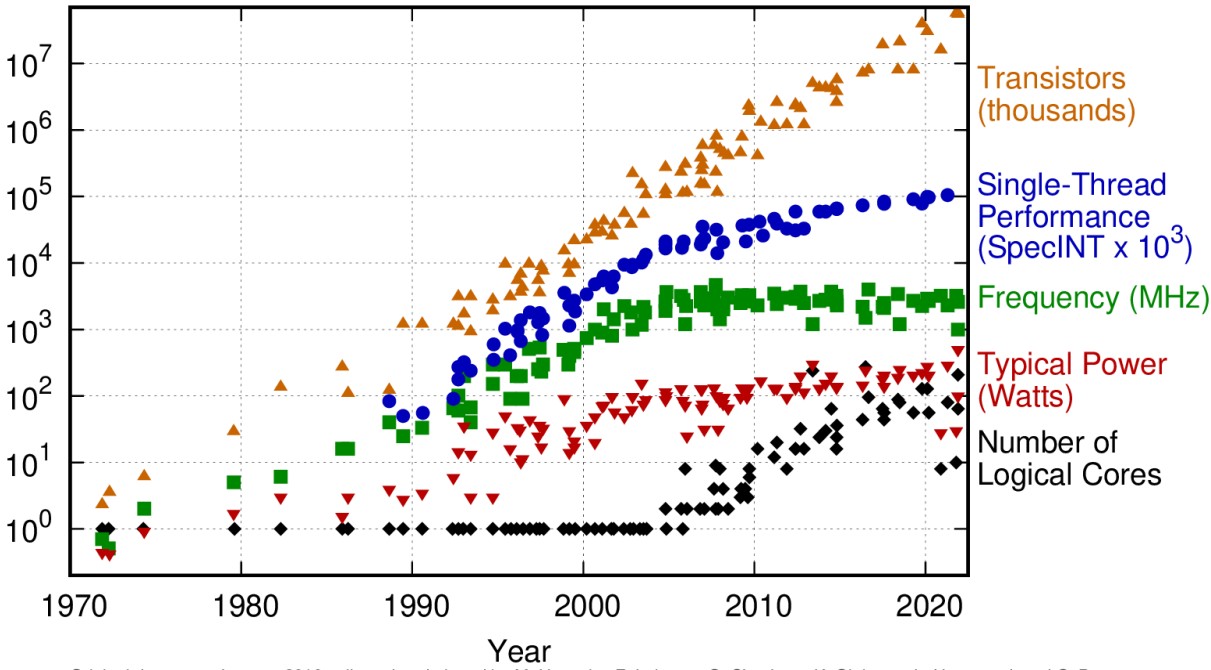

**50 Years of Microprocessor Trend Data**

Transistors (thousands)

Single-Thread Performance (SpecINT x $10^3$)

Frequency (MHz)

Typical Power (Watts)

Number of Logical Cores

Original data up to the year 2010 collected and plotted by M. Horowitz, F. Labonte, O. Shacham, K. Olukotun, L. Hammond, and C. Batten
New plot and data collected for 2010-2021 by K. Rupp

**Figure 1: 50 years of microprocessor (CPU) evolution showing the breakdown of Dennard scaling (Rupp, 2022)**

## 2 Computational Characteristics of Ocean Models

To understand why GPUs are well suited to running OOFS, it is important to consider their computational characteristics. The equations describing ocean evolution form a system of partial differential equations that are solved numerically by discretizing the model domain and then using a Finite Difference, Finite Volume or Finite Element scheme. In these forms, the bulk of the computational work takes the form of stencil computations where the update of a field at a given grid location requires that many other field values be read from neighbouring locations. This means that the limiting factor in the rate at which these computations can be done is how quickly all these values can be fetched from memory (so called 'memory bandwidth'). (Finite element schemes do have the advantage of shifting the balance in favour of doing more arithmetic operations but memory bandwidth still tends to dominate.) These computations are of course repeated across the entire model grid meaning that it is a Same Instruction Multiple Data (SIMD) problem. OOFS are therefore a very good fit for GPU architectures which naturally support massively data-parallel problems and typically provide much higher memory bandwidth than CPUs.

For execution on distributed-memory computers, OOFS typically use a geographical domain decomposition where each processor is assigned a part of the model domain. In order to handle stencil updates at the boundaries of a processor's sub-domain, it must exchange information with those processors operating on neighbouring sub-domains. Obviously, there is a cost associated with performing these exchanges which high-performance processor interconnects can only do so much to mitigate. As more processors are thrown at a problem in order to reduce the time to solution, the size of their sub-domains decreases and so too does the amount of computation that each must perform. Consequently, the relative cost of inter-processor communication becomes more significant and, after a certain point (the "strong-scaling limit"), will begin to dominate. At this point, using further processors will bring only limited performance improvements, if any.

Inter-processor communication on a GPU-based machine can be more costly as messages may have to go via the CPUs hosting the GPUs unless a machine has both hardware and software support for direct GPU-GPU communication. Communication avoiding/minimising strategies are therefore more important on these architectures. These can include algorithmic design (e.g. Silvestri et al, 2024) to allow for the overlap of communication and computation or simply the use of wider halo regions to reduce the frequency of halo exchanges.

## 3 The use of GPUs in Ocean Forecasting

Although GPUs are now a well-established HPC technology with potentially significant performance advantages for OOFS, they are not yet widely adopted in the ocean-forecasting community. For example, in Europe, NEMO (Madec et al., 2024) is the most important ocean-modeling framework; it is used operationally by Mercator Ocean International, the European Centre for Medium-Range Weather Forecasts  (ECMWF), the UK Met Office and the Euro-Mediterranean Centre on Climate Change, and other Institutes worldwide. NEMO is implemented in Fortran and parallelised with MPI and as such is limited to running on CPUs only. The German weather service (DWD) uses ICON-Ocean (Korn, 2017) which is also a Fortran model. Experiments are in progress with the use of OpenACC directives to extend this code to make use of GPUs but this functionality is not used operationally.

In the US, NOAA's Real-Time Ocean Forecast System (https://polar.ncep.noaa.gov/global/) is based on  HYCOM (HYbrid Coordinates Ocean Model, Chassignet et al., 2009). HYCOM too is a Fortran code parallelised using a combination of OpenMP and MPI. Although not used operationally, the Energy Exascale Earth System Model (E3SM) is also significant. It utilizes the MPAS (Model for Prediction Across Scales) Ocean, Sea-Ice and Land-Ice models (Ringler et al., 2013) which again is implemented in Fortran with MPI. Although a port of this was attempted through the addition of OpenACC directives, it has been abandoned due to poor GPU performance (Petersen, 2024). Instead, a new, unstructured-mesh ocean model named Omega is being developed in C++ from the ground up.  Other widely used ocean general circulation models include the  MIT General Circulation Model (MITgcm, Marshall et al., 1997) and the Modular Ocean Model, version 6 (MOM6; Adcroft et al., 2019), both of which again are  Fortran codes with support for distributed- and shared-memory parallelism on CPU.

The Japanese Meteorological Agency runs operational forecasts using the Meteorological Research Institute Community Ocean Model (MRI.COM) (Tsujino et al., 2010). As with the previous models, this too is implemented in Fortran with MPI and thus only runs on CPU.

For regional (as opposed to global) forecasts, the Rutgers Regional Ocean Modeling System (ROMS) (Shchepetkin and McWilliams, 2023) is used by centers worldwide including the Japan Fisheries Research and Education Agency, the Australian Bureau of Meteorology and the Irish Marine Institute. ROMS too is a Fortran code parallelised using either MPI or OpenMP (but not both combined) and thus is restricted to CPU execution. Although various projects have ported the code to different architectures (including the Sunway architecture for China's Tianhe machine, Liu et al., 2019), these are all standalone pieces of work that have not made it back into the main code base.

## 4 Discussion

From the preceding section, it is clear that OOFS are currently largely implemented in Fortran with no or limited support for execution on GPU devices. The problem here is that OOFS comprise of large and complex codes which typically have a lifetime of decades and are constantly being updated with new science by multiple developers. Maintainability, allowing for the fact that the majority of developers will be specialists in their scientific domain rather than in HPC, is therefore of vital importance. Given that such codes are often shared between organizations, they must also run with good performance on different types of architecture (i.e. be 'performance portable').

Previously, one generation of supercomputers looked much like the last and therefore the evolution of these computer models was not a significant problem. However, the proliferation of computer hardware (and, crucially, the programming models needed to target them) that has resulted from the breakdown of Dennard scaling has changed this (Balaji, 2021). With the average supercomputer having a lifetime of just some five years, OOFS are now facing the problem of adapting to future supercomputer architectures and this is difficult because the aims of performance, performance portability and code maintainability often conflict with each other (Lawrence et al., 2018).

*Transformation of existing codes*: To date there have been various approaches to this problem. NEMO v.5.0 (Madec et al., 2024) has adopted the PSyclone code-transformation tool (Adams et al., 2019) that enables an HPC expert to transform Fortran source code such that it may be executed on GPU using whichever programming model (i.e. OpenACC or OpenMP) is required. Previous, unpublished work found that for a low-resolution, (1 degree) global mesh, a single NVIDIA V100 GPU performed some 3.6x better than an HPC-class Intel socket. For a high-resolution, (1/12th degree) global mesh, ~90 A100 GPUs gave the same performance as ~270 Intel sockets. In both cases this is an ocean-only configuration with virtually all compute being performed on the GPUs. This is important since any computation happening on the CPU incurs substantial data-transfer costs as data is moved from the GPU to the CPU, updated, and then transferred back to the GPU. (The advent of hardware support for unified CPU/GPU memory should reduce the cost of this.) As noted earlier, ICON-O is being extended manually with OpenACC directives. There are examples of recent (i.e. experimental) models that have

moved away from Fortran in favor of higher-level programming approaches. Thetis (Kärnä et al., 2018) implements a Discontinuous Galerkin method for solving the 3D hydrostatic equations using the Firedrake framework. This permits the scientist to express their scheme in the Python implementation of Unified Form Language (Alnæs et al., 2014). The necessary code is then generated automatically. The Veros model (Häfner et al., 2021) takes a slightly different approach: its dynamical core is a direct Python translation of a Fortran code and thus retains explicit MPI parallelisation. The JAX system (http://github.com/google/jax) for Python is then used to generate performant code for both CPU and GPU. The authors report that the Python version running on 16 A100 GPUs gives the same performance as 2000 CPU cores for the Fortran version (although this comparison is slightly unfair as the CPUs used are several generations older than the GPUs).

_Performance portability tools_: Another popular approach to performance portability is to implement a model using a framework that takes care of parallel execution on a target platform. Frameworks such as Kokkos (Carter Edwards et al., 2014), SyCL and OpenMP are good examples and the new "Omega" ocean component of E3SM mentioned previously is being developed to use Kokkos. In principle this approach retains single-source science code, while enabling portability to a variety of different hardware. However, it is hard to insulate the oceanographer from the syntax of the framework (which are often only available in C++) and, while the framework may be portable, obtaining good performance often requires that it be used in a different way from one platform to another. In OpenMP for instance, the directives needed to parallelise a code for a multi-core CPU are not the same as those needed to offload code to an accelerator.

_New programming languages_: The Climate Modeling Alliance (CliMA) has adopted a radically new approach by rewriting ocean and atmospheric models from scratch using the programming language Julia (Perkel, 2019; Sridhar et al., 2022). Designed to overcome the "two-language problem" (Churavy et al., 2022), Julia is ideally suited to harness emerging HPC architectures based on GPUs (Besard et al., 2017; Bezanson et al., 2017). First results with CliMA's ocean model, Oceananigans.jl (Ramadhan et al., 2020), run on 64 NVIDIA A100 GPUs exhibit 10 Simulation Years Per Day (SYPD) at 8 km horizontal resolution (Silverstri et al., 2024). This performance is similar to current-generation CPU-based ocean climate models run at much coarser resolution (order 25-50 km resolution). Similarly promising benchmarks have been obtained with a barotropic configuration of a prototype of the MPAS-Ocean model, rewritten in Julia (Bishnu et al., 2023). Such performance gains hold great promise for accelerating operational ocean prediction at high spatial resolution run on emerging HPC hardware.

_Toward energy efficient simulations_: Increased resolution, process representation, and data intensity in ocean and climate modeling is vastly expanding the need for compute cycles (more cores and smaller time steps). As a result, the ocean, atmosphere, and climate modeling community has recognized the need for their simulations to become more energy efficient and reduce their carbon footprint (Loft, 2020; Acosta et al., 2024; Voosen, 2024). Owing to their architecture, GPUs can play a significant role in reducing energy requirements. A related research frontier being spearheaded by the atmospheric modeling community is the use of mixed or reduced precision to speed up simulations (Freytag et al., 2022; Klöwer et al, 2022; Paxton et al., 2022), with a potentially desirable side effect of natively capturing stochastic parameterizations

(Kimpson et al., 2023). GPUs are ideally suited for such approaches, but successful implementation depends heavily on the model's numerical algorithms.

*Data-driven operational ocean forecasting*: Operational weather and ocean forecasting are facing the potential of a paradigm shift with the advent of powerful, purely data-driven methods. The numerical weather prediction (NWP) community has spearheaded the development of machine learning-based emulators that perform several orders of magnitudes faster than physics-based models (e.g., Bouallègue et al., 2024; Rasp et al., 2024). Such emulators have the potential to revolutionize probabilistic forecasting and uncertainty quantification, among others. The computational patterns underlying the ML algorithms, such as parallel matrix multiplication, are ideally suited for general-purpose GPU architectures. Whereas these methods have been driven to a large extent by private sector entities and require access to increasingly large GPU-based HPC systems for training, corresponding efforts in operational ocean forecasting are only now beginning to catch up. A review of the rapidly changing landscape of AI methods in the context of ocean forecasting is attempted in Heimbach et al. (2024).

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

## Competing interests

Author A. Porter has declared that he is an author of the PSyclone package and a co-chair of the NEMO HPC Working Group.

## Data and/or code availability

No data or code is associated with this work.

## Authors contribution

AP created the first draft of this work. PH assisted with updating the text in the light of the reviews received.

## Acknowledgements

The authors would like to thank the reviewers for bringing the E3SM work to their attention.