# Peer review of "Unlocking the Power of Parallel Computing: GPU technologies for"

_State of the Planet, 2024_

## Referee Comment (RC2)

Review of "Unlocking the Power of Parallel Computing: GPU technologies for Ocean Forecasting" by Porter and Heimbach

This review article covers an important topic on the transition in supercomputing architectures and its ramifications for ocean modelers. It is relevant today to all computational physics models, and the article describes the difficulties experienced by climate and weather centers, and their plans to deal with the transition from CPU to GPU-based machines.

I am one of the lead developers of the ocean component for the US DOE Energy Exascale Earth System Model. We went as far as we could to enable GPU usage in the Fortran-based MPAS-Ocean model using OpenACC in addition to the MPI and OpenMP. In the end we were not able to achieve the GPU throughput needed with Fortran-based OpenACC. We are now writing a new unstructured-mesh ocean model, Omega, in C++ from the ground up. Our primary motivation is to move the E3SM ocean component fully to GPUs in order to use the large computing resources available in the DOE. The Kokkos performance portability library is central to this plan, as we can take advantage of specialists who tune this intermediate layer to new and upcoming architectures. As you say in the text, Kokkos requires that we move from Fortran to C++. We see this code rewrite as a large, one-time cost to transition to a modern coding language and standards so that we are more adaptable and resilient to new hardware designs for many decades to come.

The authors are free to use the information or a quote from the previous paragraph if they would like, citing simply "written communication". We are currently writing a manuscript documenting Omega development. At this time one may see our online documentation and abstracts from our conference presentations:
- Omega documentation:
  https://docs.e3sm.org/Omega/omega/develop/design/OmegaV0ShallowWater.html
- AGU Ocean Sciences 2024:
  https://agu.confex.com/agu/OSM24/prelim.cgi/Paper/1482970
- AMS 2025
  https://ams.confex.com/ams/105ANNUAL/meetingapp.cgi/Paper/453456

Note that the second abstract is in the session "GPU Accelerated Computing for Weather, Water, and Climate", which is relevant to this publication and may provide you with further information.

The publication is in line with my understanding of the current state of the field of OOFS as it transitions to GPU-based architectures. I appreciate the author's summary of the state of the science among many modeling groups.

I think that the question of how models will change to incorporate AI/ML will cause much deeper disruption in computational physics than the current changes to GPU architectures. In our case, we are rewriting our ocean model with the same algorithms but new performance libraries. The AI/ML revolution is causing much greater uncertainty, as it

may displace the numerical methods we've used for 70 years. I think this point deserves a few more sentences in the text.

Line 43: This is a good time to update the top 500 reference to the most recent one, from November 2024. The top three on the list are by the US DOE and are all hybrid CPU/GPU machines with significant GPU components.

L 59: It is important to say that computational fluid dynamics models are always constrained by communication on multi-node distributed memory systems. Advection-diffusion equations transfer information across spatial cells, so this is inescapable. The CPU to GPU memory transfer now adds another layer of communication time. There are a few ways to reduce these costs. One is to overlay non-conflicting computation during the communications step with carefully designed algorithms. See your ref. Silvestri et al. 2024 for a great example of this with the baroclinic/barotropic splitting. An additional method is to take advantage of specific architectural features, like direct GPU-to-GPU communication.

Lines 98-112: It would be helpful to have more statistics about the reported success of the OpenACC approach in ocean models. If it is possible to add more numbers documenting the percent of compute that was transferred from CPU to GPU, and the speed-up times for these different models, that would be insightful.

The Strauss et al. 2023 reference is now:
Bishnu, S., Strauss, R. R., and Petersen, M. R.: Comparing the Performance of Julia on CPUs versus GPUs and Julia-MPI versus Fortran-MPI: a case study with MPAS-Ocean (Version 7.1), Geosci. Model Dev., 16, 5539–5559, https://doi.org/10.5194/gmd-16-5539-2023, 2023.

Sincerely,
Mark R. Petersen
Los Alamos National Laboratory

---

## Author Response (AR1)

RC2:

Many thanks for the useful review and the additional technical information on the ocean component of E3SM that is being developed. It will be very interesting to see how your work there progresses. We have added this information to the text.

The reviewer comments:

*The AI/ML revolution is causing much greater uncertainty, as it may displace the numerical methods we've used for 70 years. I think this point deserves a few more sentences in the text.*

This is a good suggestion. We've added some text about this issue.

To respond to the remaining points:

L43: We have updated the Top500 reference (and associated text) to use the latest list as of Nov 2024;

L59: Thanks for pointing out that we'd missed describing inter-processor communication and the additional complexities this brings for GPUs. We've added a paragraph explaining the need for this communication and the implications for GPU.

L98-112: We've added a little more detail here.

The Strauss et al. reference has been updated (thanks).

–

RC1:

Many thanks for the review. Below we address the reviewer's comments and suggestions (reviewer's comments in red, our response in black):

*This difficulty - and possible solutions - should be discussed in the paper and would increase greatly the value of the contribution. These solutions range from using software other than MPI, defining a new interface (with middleware) between FORTRAN codes and parallel/distributed processing at both node and processor levels or revising the FORTRAN codes. Developments in processor technology may introduce other opportunities beyond the current generation of GPUs.*

We concur with the reviewer regarding challenges and solutions, however we think that the manuscript does describe various solutions, i.e., we address the various ways in which OOFS can make use of GPUs, ranging from the addition of OpenACC/OpenMP directives to existing

Fortran through to complete re-writes in C++, Python and Julia. We are unsure in what way we should extend the discussion.

*Also, it would appear that the problem is being addressed by OpenMPI and that they have a version of MPI that takes advantage of  GPUs.*

As you say, GPU-aware MPI implementations are important and we have extended the text to make this clear.